# Fiscal Deficit and Its Impact on Economic Growth: Evidence from Bangladesh

**Mohammed Ershad Hussain [1],* and Mahfuzul Haque [2]**

1    Department of Economics and Finance, Dillard University, New Orleans, LA 70122, USA
2    Scott College of Business, Indiana State University, Terre Haute, IN 47809, USA;
     mahfuzul.haque@indstate.edu
*    Correspondance: mhussain@dillard.edu

**Abstract:** The findings from the VECM for BBS data reveal that there is a positive and significant relationship between FD and GDPGR, supporting the Keynesian theory, while findings from the VECM for World Bank data indicate that the impact of Fiscal Deficit (FD) on GDPGR is mild but negative and significant at the 5% level. This contradicts the Keynesian theory, but is in accord with Neo-classical theory which asserts that fiscal deficits lead to a drop in the GDP. Nevertheless, the government must strive to keep deficit under control, not to hamper growth, and expenditure ought to be set so as to avoid massive deficits leading to debt financing and the crowding-out effect of private investment. If deficits become unsustainable, it can lead to higher interest payments, and the government may well default. Although in the economic literature, there is no definitive conclusion as to whether fiscal deficit helps or hinders economic growth for any country, many argue that fiscal deficit leads to economic growth of a country, which cannot be achieved only through domestic savings, not enough for investment. It can be assumed safely that to some extent fiscal deficit is good for economic growth if the borrowed money is spent on beneficial projects, provided the return from such investments exceeds the funding cost. For future research work, it will be interesting to examine the relationships between government spending, economic growth and long-term interest rate for Bangladesh.

**Keywords:** Fiscal Deficit; GDP; Vector Error Correction Model; Augmented Dickey Fuller and Bangladesh

**JEL Classification:** E62, F43, H61, H62, O40

## 1. Introduction

Bangladesh economy has drawn increased international attention in recent times. Many articles have been written eulogizing the country's economic success story. Business Insider of the UK terms Bangladesh as the "New Asian Tiger". According to their published report, its economy has been one of the top performers in Asia over the past decade, averaging an annual growth of more than 6%, with most of the growth that the country has achieved coming from the garment sector. The report also states that if Bangladesh is to achieve the government's ambitious growth target of 8% a year by 2020, it is essential that it starts to diversify out of the garment sector into other sectors, such as electronics and other consumer durables, where there is more scope to add value. Bangladesh's public equity valuations are also beginning to catch up with its high growth and high returns on capital.[1]

---

1    Source: http://www.businessinsider.com/bangladesh-is-the-new-asian-tiger-2017-4

Asian Development Bank's recent report on the Bangladesh economy states "Bangladesh's GDP growth accelerated to 7.1% in 2016 from 6.6% in 2015."[2] The market-based economy of Bangladesh[3] is the 44th largest in the world in nominal terms, and 32nd largest by purchasing power parity; it is classified among the Next Eleven (N-11) emerging market economies by Goldman Sachs.[4] The financial crisis of 2007–2008 in the United States was followed by a global economic meltdown that led many to believe that the Bangladesh economy too will be adversely affected. However, Bangladesh came out doing fairly well. According to the study by Ali et al. (2011), the authors find that the macro economy of Bangladesh has shown remarkable resilience in the face of this massive global crisis, and the impact has been minimal and limited to a moderate slowdown of the economy[5].

Here in this study, our purpose is to analyze the relationship between Fiscal Deficit and Economic Growth in Bangladesh, employing two separate datasets, a longer time-series from the Bangladesh Bureau of Statistics (BBS)[6] for the period 1993–1994 to 2015–2016 for Fiscal Deficit (FD) and GDP in Current Billion Taka and adjusted for inflation with the National Consumer Price Index (CPI). This is the main price index published by the government to measure inflation. The second dataset, which is shorter than the first one, is from the World Bank Development Indicators (WBDI)[7] for Fiscal Deficit (FD) in Million Taka and annual growth rate of GDP (GDPGR) for Bangladesh over the period 2001–2014. For both datasets, we conduct the Augmented Dickey–Fuller, Phillips–Parron and Kwiatkowski–Phillips–Schmidt–Shin (KPSS) tests and rank tests to fit a Vector Error Correction Model (VECM). We also conduct stability tests, tests of autocorrelation and heteroskedasticity to check for reliability of result. In prelude to our study, estimation results from the BBS data show that there is indeed a long-run positive relationship between the two variables. This finding supports Keynesian argument which recommends that government expenditure will result in higher economic growth, but contrary to the Monetarist beliefs that fiscal policy does not matter.

Our study is different from previous study in several key areas: we use two datasets from two sources; one is the official source of the Bangladesh Government and the other is the World Bank online database. We collected the longest series that was available to us. Another key feature covered in the present study is that we have used a series of tests in every stage, often checking the result of one test with another. For example, we conduct three different approaches to unit-root tests, Dicky–Fuller tests, Phillips–Perron tests, and KPSS tests. We also made use of the lambda max statistics and lambda trace statistics for the rank tests. Several methodologies were used to conduct auto-correlation and heteroskedasticity. Two alternative methods of stability tests were performed: Unit-root circle

analysis of stability of the system, the other, cusum square bound tests of stability. Breusch–Pagan test and White test of heteroskedasticity were done also. Another distinguishing feature of the paper is that it recommends policy suggestions to government and non-government / donor agencies by substantiating that government budget deficit has an important impact on economic growth.

[2]  Source: https://www.adb.org/countries/bangladesh/economy
[3]  According to World Bank's report "Bangladesh economy remained strong and resilient despite external and internal challenges. Bangladesh is among the top 12 developing countries with a population of over 20 million, who achieved 6 plus percent growth in 2016. By any standards, Bangladesh economy has done well. Bangladesh needs to focus on a growth agenda centered on sustainable and inclusive growth. Overall fiscal deficit was contained at 3.6 percent of GDP, despite low tax collections due to political disruptions, and a shift towards low duty commodity import. NBR revenue collection fell short, at Tk. 1208.2 billion in FY15 compared to Tk. 1497.2 billion target. The Annual Development Program (ADP) implementation lags behind with an out turn of Taka 685 billion, though improved from last year. Domestic financing of budget deficit increased to 1.4 percent of GDP in FY15 from 1.1 percent in FY14. This came entirely from 93 percent increase in non-bank borrowing". Source: https://www.worldbank.org/en/news/feature/2016/04/30/bangladesh-development-update-bangladesh-economy-requires-focus-on-sustainable-and-inclusive-growth-moving-forward
[4]  Source: https://en.wikipedia.org/wiki/Economy_of_Bangladesh
[5]  According to a report published by World Bank (2017), "The Bangladesh economy is weathering persistent global uncertainties relatively well. Growth remained resilient, aided recently by recovery in exports and private investments. Inflation has decelerated benefitting from soft international commodity prices and prudent macroeconomic management. The outlook for Bangladesh's main export destinations is projected to improve, although downside risks remain elevated".
[6]  Bangladesh Bureau of Statistics (BBS) is only the national Statistical institution responsible for collecting, compiling and disseminating statistical data of all the sectors of the Bangladesh economy to meet and provide the data-needs of the users and other stake holders like national level planners and other agencies of the Government. The role of the BBS in providing necessary statistics for preparing the various national plans and policies for the overall development of the country is very significant.
[7]  World Bank Development Indicators is the primary World Bank collection of development indicators, compiled from officially-recognized international sources. It presents the most current and accurate global development data available, and includes national, regional and global estimates.

This paper is structured as follows. Section 2 covers the literature review of the major works in the field, followed by a Section 3 which deals with the data source and methodology. Section 4, discusses the results and Section 5 presents the conclusions of the paper.

## 2. Literature Review

Does fiscal deficit really impact economic growth? This has been an argumentative topic debated widely in the economic literature without reaching any conclusive outcome. Fiscal deficit is stated to be one of the significant variables to have a sway on growth. Some different schools of thought exist in the literature on how to relate the fiscal deficit and economic growth. According to the Keynesian economics, government expenditure is one important component of Aggregate Demand (AD) in the economy. Whenever AD falls short (during recessions), the government can increase expenditure, which in turn will increase AD, and in turn, will stimulate the economy. This solution based on government stimulus worked well to increase output, employment and income that brought the U.S. economy out of the Great Depression of 1929–1933 and during the most recent Great Recession of 2007–2009. The same practice was followed by several other countries over the years to stimulate AD and the pace of economic growth.[8]

The classical economists believe that government stimulus financed by domestic debt will result in an increase in government investment at the cost of private investment. Interest rates will increase when governments borrow from the domestic market, which will cause a decline in consumption and private investment. This is the popular "crowding-out" argument against government spending. Massive government spending through borrowing may lead to "crowding-out" effect, but the protagonist of fiscal deficits sees the other side of the coin. They counter the "crowding-out" assertion agreeing to the fact that though crowding out is possible in financial markets, there is a converse effect as well. The conjecture, known as "crowding in", contends that government spending will create an increase in aggregate demand. As the economy expands, the private sector has to bump up production and businesses find it profitable to add to their capacity so as to meet the greater consumer demands, thus more production requires that additional capital be invested. Some economists say that the myth in the crowding-out argument lies in believing that the economy's flow of saving is actually fixed. If government deficits succeed in raising output, then more income and, hence, more saving can take place, thus *both* government *and* industry can borrow more.

Another school of thought is the Ricardian Equivalence, which postulates that fiscal deficit cannot stimulate the economy. If the agents are rational then they will see that the increased deficit implies future taxes of which the present value equals the value of the deficit. Thus, they will act as if the deficits do not exist, which means that consumers and investors will ignore the stimulus. Rostow's Stages of Growth model is one of the most influential development theories of the twentieth century. He studied the history of economic development in developed countries (US and Western Europe) and identified many stages of economic development. Five stages of development have been labeled that every country must pass through as they develop from the initial state (Traditional Society) to an advanced stage (The Age of High Mass Consumption). Government investment is necessary to develop the

---

[8] Briotti (2005) reviews the theoretical and empirical literature that has investigated the conditions under which a contractionary fiscal policy is effective in reducing debt and deficit, but does not have a negative effect on growth. The author concludes that the theoretical impact of fiscal policy on aggregate demand and economic activity depends largely on the conceptual framework considered and its assumptions about the world. Empirical studies based on macro-econometric model simulations find evidence that fiscal consolidations lead initially to production losses, while they can result in a higher output in the medium term. Empirical studies focusing on episodes of changes in fiscal policies provide in turn evidence that under certain circumstances austerity measures may have an expansionary impact on the economy.

infrastructure and social capital that will open up the way for the private sector to come forward to invest and develop. Thus, government investment in the key sectors of the economy should work as a facilitator for the growth of the entire economy. According to the Unbalanced Growth Theory of Hirschman, "Development is a chain of disequilibria that must be kept alive rather than eliminate the disequilibrium in which profits and losses are symptoms in a competitive economy." If the government invests in key sectors of the economy and builds the necessary infrastructure, then private sector will come forward and invest in the other sectors that become profitable once the initial government investments are made. Developing counties often have low savings and investment compared to more developed countries. Thus, the developing countries often do not have enough resources to invest in all sectors. Therefore, limited investment in key areas is a very attractive strategy to break the 'vicious circle of poverty'. Hirschman pointed out that "shortages created by unbalanced growth offer considerable incentives for inventions and innovations. Imbalances give incentive for intense economic activity and push economic progress." (Hirschman 1969). Haider et al. (2016) studied the relationship between budget deficit (several other variables) and economic growth for Bangladesh. The authors employ different statistical tests and models (i.e., Unit-root test, VAR, VEC, Granger causality) to find out the impact of budget deficit on GDP growth. Using quarterly data over the period 2000–2012, their findings from the research imply that there are cointegrating relationships among budget deficit, inflation and exchange rate and there is a negative impact of budget deficit on GDP growth. Ebney Ayaj Rana and Abu N.M. Wahid (Rana and Wahid 2016) conducted a time-series analysis using ordinary least squares estimation, vector error correction model, and Granger causality test. Their findings suggest that the government budget deficit has a statistically significant negative impact on economic growth in Bangladesh. Policy implications of findings include reestablishing the rule of law, political stability in the country, restructuring tax structure, closing tax loopholes, and harmonizing fiscal policy with monetary policy to attract additional domestic and foreign investment. Hussain and Haque (2017) studied the relationship between money supply and GDP growth in Bangladesh. The paper mainly employs two econometric models to achieve the empirical results: the first one under the Engle and Granger (1987). The second model utilizes the cointegration procedure and the associated Error Correction Model (ECM) framework to study the relationship between these variables and find that money supply has an important impact on the growth rate of output in the long run for Bangladesh. The authors suggested that the government should maintain consistency in formulating monetary policy and follow a rule instead of discretion, such as the "Taylor rule" to allow money supply to increase at a steady rate keeping pace with the growth of the economy. Hussain and Haque (2016) also studied the relationship between FDI, Export and Economic Growth in Bangladesh using the Vector Error Correction Model (VECM), and their findings reveal that there is a relationship between foreign direct investments, trade, and growth rate of per capita GDP. As the FDI and trade are two important components of economic growth in Bangladesh, it is important to frame policies that promote growth and reduce the barriers for capital flows. Nayab (2015) study examines the impact of budget deficit on economic growth in Pakistan during the period from 1976–2007. Cointegration technique, VAR Granger causality test and vector error correction model are used. Findings indicate that there is a positive impact of budget deficit on economic growth, which is significant. The author claimed that the results of the study also support Keynesian view about budget deficit. Hassan et al. (2014) studied the effect of Government Deficit spending on the GDP in the United States. These authors analyzed data from the US and developed a time series model showing the relationship between deficit spending and GDP. The SAS software was used in the data analysis. The Johansen cointegration analysis was performed in order to determine if cointegration exists between GDP and deficit spending, unemployment rate, interest rate, and inflation rate. Results revealed that government deficit spending had a negative effect on GDP. Only unemployment had a negative effect on the GDP in the presence of deficit spending. It is interesting to note that GDP was integrated (having a long-run equilibrium relationship) with unemployment rate, interest rate, and inflation rate. Krugman (2012) argued that deficit spending did not help the economy in the recent recession, because it was not enough to cause

an increase in demand and economic growth. If Krugman's argument is correct, then the observed negative relationship between deficit spending and GDP may not indicate cause and effect. If deficit spending has a negative effect, it would be because of its effect on raising the long-term interest rate. Navaratnam and Mayandy (2016) seek to examine the impact of fiscal deficit on economic growth in some selected South Asian countries, namely, Bangladesh, India, Nepal, Pakistan and Sri Lanka, using time series annual data over the period 1980–2014. Their study employs the econometric techniques of cointegration and Granger causality test to examine the dynamic relationship among the selected variables. The results from their study confirm that the fiscal deficit has a negative impact on economic growth in the South Asian countries considered in this study except Nepal, which confirmed the positive impact. The results also highlighted that the direction of causality for the South Asian Association for Regional Cooperation (SAARC) countries is mixed where fiscal deficit causes economic growth for Bangladesh, Nepal and Pakistan, but the reverse is true for India and Sri Lanka. Hassan and Akhter (2014) studied the relationship between budget deficit and economic growth with the help of a Shojai (1999) model. They focused on relationships between Gross Domestic Product (GDP) regressed with inflation rate, real interest rate, real effective exchange rate, budget deficit and gross investment with a sample for the period 1976–1977 to 2011–2012. The authors used augmented Dickey–Fuller (ADF) and Johansen Cointegration and in the second step used the Vector Error Correction Model (VECM). The empirical results find statistically significant negative effect of budget deficit over economic growth of Bangladesh. Ramu and Gayithri (2016) aim of the study is to examine both the short-run and long-run relationship between fiscal deficit and economic growth in India by covering the time period from 1970–1971 to 2011–2012 by Johansen cointegration test, Granger causality test, and Vector Error Correction Model (VECM) technique. The study finds that fiscal deficit adversely affects GDP supporting the mainstream neo-classical theory. They state that when fiscal deficit is bifurcated into effective fiscal deficit and revenue deficit, the former has a significant positive relation whereas the latter has a negative relation with GDP. Their result argues for reducing the revenue deficit part in the fiscal deficit. The effective fiscal deficit on the one hand enhances capital formation directly and, on the other hand, indirectly encourages the private sector to invest more. Government investment in infrastructure will have a crowding-in effect for private investment. The control variables used in the paper, like private investment, indicated a positive relation with GDP while the exchange rate and tax revenue, with some exception, proved to be having a negative relation with GDP as expected. They support the 'Golden Rule' of public finance and argue that the fiscal deficit amount should be used for capital formation purpose and not for the current consumption of the Government. Cinar et al. (2014) examined the best five (5) and worst five (5) countries in the Eurozone according to their debt ratios and discussed the growth rates, debt ratios and budget deficit variables of these countries with special focus on the period 2008 and the recent Great Recession of 2007–2009. The authors used a Panel ARDL model for the 2000Q1–2011Q4 period. Authors found that the analysis results showed that conjectural deficit policy (functional fiscal policy) had a positive effect on economic growth in the short run. Taylor et al. (2012) examined the impact of the 'primary' fiscal deficit on economic growth and debt via a four-dimension VAR and several specifications of Impulse Response Function (IRF). The authors reviewed the last 50 (1961–2011) years' data and presented a vector auto-regression (VAR) analysis of dynamic interactions among fiscal deficits, economic growth and interest rates. The authors found that higher spending and lower taxes have a positive impact on output expansion. Odhiambo et al. (2013) studied the impact of fiscal deficit on economic growth in Kenya over the period 1970–2007. The authors used Dickey–Fuller and ADF tests and Johansen cointegration test and found there was a positive impact of budget deficit on economic growth. They concluded that government should undertake prudent financial management and be careful not to 'crowd out' private investment. Mohanty (2012) examined the short- and long-run relationship between fiscal deficit and economic growth in India from 1970 to 2012. The study found a negative and significant relationship between fiscal deficits and economic growth in the long run. The short-run results discard the relationship between fiscal deficits and economic growth. The findings also revealed

that the negative impact of post-reform fiscal deficit on economic growth is more than the impact of pre-reform's fiscal deficit. Fatima et al. (2011) examined the impact of fiscal deficit on investment and economic growth using a time series for the period 1980–2009 for Pakistan within the framework of 2SLS base on a simultaneous equations framework. Authors found that fiscal deficit affects economic growth directly and the persistent balance of payment deficit is linked to fiscal deficit. Bhoir and Dayre (2015) examined the impact of fiscal deficit on economic growth for the Indian economy for the period 1991–1992 to 2013–2014. These authors used the ordinary least squares method and found no significant relationship between the two variables. The above authors suggested that the government of India should instead focus on human development indicators, such as health, education and infrastructure development to enhance the productivity of human and physical capital that in turn will accelerate economic growth. Edame and Okoi (2015) examined the impact of fiscal deficit, interest rate and gross fixed capital formation on economic growth in Nigeria during the periods 1985–1998 and 1999–2013, where the former was labelled as military regime and the latter as the democratic regime. Authors used the ADF and the Chow endogenous breaks tests to examine the growth impact of fiscal deficits before and after the advent of democracy in Nigeria in 1999. Authors found that fiscal deficit had a significant impact on economic growth during the military regime only. They also found that the interest rate did not have a significant growth impact during both the regimes, while the gross fixed capital formation had a significant growth impact during both regimes. Alfonso and Turin (2008) analyzed both the long- and the short-run relationship between government expenditure and potential output in EU countries by means of pooled mean group estimation (Pesaran et al. 1999) over a sample comprising EU-15 countries during the 1970–2003 period. The authors used the Pooled Mean Group (PMG) estimator, which permitted them to combine the precision of the estimates allowed by pooling the data across the cross-country dimension. Estimation results showed that they cannot reject the hypothesis of a common long-term elasticity between cyclically adjusted primary expenditure and potential output close to unity. However, the catching-up countries, in fast-aging countries, in low-debt countries, and in countries with weak numerical rules for the control of government spending, the elasticity was significantly higher than unity. Iya et al. (2014) examined the impact of fiscal deficit on the Nigerian economy during the period 1981–2009. Authors found that there was a one-way causation between exchange rate and real GDP. The results of OLS also indicated that interest rate, exchange rate, and government fiscal deficit had a positive impact on economic growth in Nigeria during the period under consideration. Vamvoukas (2000) in a related study examined the relationship between fiscal deficits and interest rate for Greece over the period 1970–1994. They followed the Barro (1981, 1987) approach and divided government spending into permanent and transitory components and employed an ECM framework. Based on the results of the model, they found strong support for the existence of short- and long-term relationships between interest rate and budget deficit and support for the Keynesian proposition. Martin and Fardmanesh(1990) paper examined the impact of the key fiscal variables—taxes, expenditures, and deficits—on economic growth performance, using a reduced-form model and cross-sectional data for a sample of 76 developed and developing countries for the period 1972–1981. Its simultaneous consideration of fiscal variables overturns the results of some existing studies. While taxes seem negatively associated with GDP growth, they are concomitant with a higher rate of growth when their benefits in terms of reducing deficits are considered. The positive association of government expenditures with GDP growth is rendered negative when their impact on deficits is factored in. Deficits are contractionary, and deficit-reducing tax increases and expenditure cuts are positively associated with growth. A balanced budget expansion of taxes and expenditures was negatively associated with growth. When separating the sample into low-, middle- and high-income countries, these results hold only for the second group, indicating that the level of development influences the linkages between fiscal variables and GDP growth.

### 3. Data and Methodology

In this study, we use data from two sources, the Bangladesh Bureau of Statistics (BBS Bangladesh Bureau of Statistics), and the World Banks's World Development Indicators (WDI 2017, https://data.worldbank.org/data-catalog/world-development-indicators). The BBS dataset is the local source and it is the only national institution in Bangladesh responsible for compiling and disseminating statistical data. On the other hand, the World Bank dataset is the foreign source. World Bank Development Indicators is the primary World Bank collection of development indicators, compiled from officially recognized international sources. To carry out statistical analysis, we collected a longer series dataset from 1993–1994 to 2015–2016 for GDP and Fiscal Deficit in Billion Taka adjusted for inflation, which comes from the Bangladesh Bureau of Statistics (BBS), while the second dataset, the annual growth rate of GDP (GDPGR) and Fiscal Deficit (FD) in Million Local Currency data, comes from the World Bank dataset. For sake of brevity, tables from the WB dataset are presented in the appendix. In the first step of our analysis, we do the ADF tests to test for stationary or not stationary. Then, we proceed to find out the lag order and then do the rank test for VECM. Based on the lag selection tests, we fit the following form of Augmented Dickey–Fuller test with a constant and a trend, where $y$ is the series

$$\Delta y_t = \alpha + \gamma y_{t-1} + \lambda_t + \sum_{s=1}^{m} a_s \Delta y_{t-s} + v_t$$

We test whether or not the data series have a unit root with the help of this ADF test.

Next, we test if there is a long-run cointegration relation among the non-stationary data or not. Engle and Granger (1987) established the method of testing the cointegration, which has several limitations. The Fully Modified Ordinary Least Squares (FM-OLS) approach for cointegration was developed later on by Phillips and Hansen (1990), which considered the problems of correction factors of endogeneity and serial correlation problems. FM-OLS is used in panel data. In the present paper, we use the Johansen and Juselius (1990) system method of cointegration to solve the problems of bi-directional causality and to treat all variables as endogenous to the system. Johansen method has two tests to check cointegration, namely the $lamda_{trace}$ and $lamda_{max}$. In the $lamda_{trace}$ statistic test, it is held that the null hypothesis in which the number of distinct cointegrating vectors is less than or equal to $r$ against a general alternative of $k$ cointegration relation ($k$ is the number of endogenous variables in the system. On the other hand, the $lamda_{max}$ statistic tests the null that the number of cointegrating vector is $r$ against the alternative $r + 1$ cointegrating vectors. In this paper, we use the first test. Once we confirm the existence of cointegrating relation, we proceed with the VECM model estimation.

In the error correction model, we specify the following model,

$$\Delta GDP\ Growth\ Rate_t = \alpha_{10} + \sum_{i=1}^{l11} \alpha_{11i}\Delta Fiscal\ Deficit_{t-i} + \sum_{j=1}^{l12} \alpha_{12j}\Delta GDP\ Growth\ Rate_{t-j} - \gamma_{13}ECM_{t-1} + \varepsilon_{1t}$$

$$\Delta Fiscal\ Deficit_t = \alpha_{20} + \sum_{i=1}^{l21} \alpha_{21}\Delta Fiscal\ Deficit_{t-i} + \sum_{j=1}^{l22} \alpha_{22}\Delta GDP\ Growth\ Rate_{t-j} - \gamma_{23}ECM_{t-1} + \varepsilon_{2t}$$

Here $\Delta$ stands for the first difference operator, $\gamma_1$ and $\gamma_2$ are error correction terms, and $\varepsilon_{1t}$ and $\varepsilon_{2t}$ are random error terms. while $s$ and $m$ are the number of lag lengths. The error correction terms show adjustment towards long-run equilibrium and the $\alpha_1$ and $\alpha_2$ coefficients show the adjustment in short-term equilibrium.

### 4. Discussion and Results

We present the analysis of fiscal deficit (RFD) and GDP (RGD) (both adjusted for inflation) based on the BBS series of 1993/94 to 2015/16. Table 1 presents the summary statistics of the two variables. The mean real GDP (RGDP) during this period was 3232.125 Billion Taka and Fiscal Deficit (RFD) was 140.291 Billion Taka. Table 2 shows the correlation coefficient between the variables, which is −0.9804.

**Table 1.** Summary Statistics (Period: 1993/94-2015/16).

|        | Obs. | Mean     | Std. Dev. |
| ------ | ---- | -------- | --------- |
| RGDP   | 23   | 3132.125 | 2098.388  |
| RFD    | 23   | −140.491 | 100.771   |

**Table 2.** Correlation Coefficient.

|        | RGDP    | RFD |
| ------ | ------- | --- |
| RGDP   | 1       |     |
| RFD    | −0.9804 | 1   |

Table 3 Panel A presents the Augmented Dickey–Fuller tests of the two series in level and in first difference. The Null hypothesis in all variables is that the series has a unit root. We find that the test statistic for RGDP is 1.39 which is larger than the critical values of the test statistics at all the 1%, 5%, and 10% levels of significance. Therefore, we find that this series has unit root, that is, the series is not stationary at levels. We test for first difference and find that the value of test statistics is −4.49, which is smaller than the critical values 5% and 10% level of significance. Thus, the first difference of RGDP series does not have a unit root, that is, the series is stationary. Similarly, RFD is not stationary at levels (with test statistics of 1.50) but the first difference does not have unit root for 5% and 10% level of significance (the value of the test statistics is −4.29 which is smaller than the critical value of the test statistics at all the 1%, 5% and 10% levels. In Panels B and C of Table 3, we present results for Phillips–Perron and KPSS tests, respectively. We find similar results in Panels A and B. In KPSS test, the null hypothesis is that the series is trend stationary. At the 1% level, critical value is 0.216. RGDP is trend stationary in both level and first difference. However, RFD is trend stationary in first difference only.

In Table 4, we present the result for lag selection. We find that several selection criteria, including AIC and SBIC, all indicate to lag one. We follow this when we measure the VECM. In Panel A of Table 5, we present the result of rank test ($lamda_{trace}$) and find that the system has a rank of one. In the rank test, if the trace statistics corresponding to a specified rank are greater than the critical value, we reject null hypothesis, which postulates that the number of co-integrating relationship is equal to $r$ (given in the maximum rank column of the output). In the test of $r = 0$, null hypothesis is rejected. In the case of $r = 1$, it is not rejected, that is, the value of the lambda trace statistics is 0.708 which is smaller than the 5% critical value of 3.84. In Panel B of Table 5, we present the result of Max statistics $lamda_{max}$ and find similar results.

**Table 3.** Unit-Root Tests.

|  | Test Statistics | 1% Critical | 5% Critical | 10% Critical |
|---|---|---|---|---|
| **Panel A: Dickey–Fuller Tests** | | | | |
| RGDP | | | | |
| Z(t) | 1.39 | −3.75 | −3.00 | −2.63 |
| D.RGDP | | | | |
| Z(t) | −4.49 | −3.75 | −3.00 | −2.63 |
| RFD | | | | |
| Z(t) | 1.50 | −3.75 | −3.00 | −2.63 |
| D.RFD | | | | |
| Z(t) | −4.29 | −3.75 | −3.00 | −2.63 |
| **Panel B: Phillips–Perron Tests** | | | | |
| RGDP | | | | |
| Z(rho) | −7.29 | −17.20 | −12.50 | −10.20 |
| Z(t) | −2.13 | −3.75 | −3.00 | −2.63 |
| D.RGDP | | | | |
| Z(rho) | −19.32 | −17.20 | −12.50 | −10.20 |
| Z(t) | −5.17 | −3.75 | −3.00 | −2.63 |
| RFD | | | | |
| Z(rho) | 4.39 | −17.20 | −12.50 | −10.20 |
| Z(t) | 5.90 | −3.75 | −3.00 | −2.63 |
| D.RFD | | | | |
| Z(rho) | −19.76 | −17.20 | −12.50 | −10.20 |
| Z(t) | −3.86 | −3.75 | −3.00 | −2.63 |
| **Panel C: KPSS Tests** | | | | |

| Lag Order | Test Statistic RGDP | | Test Statistic RFD | |
|---|---|---|---|---|
| 0 | 0.070 | | 0.457 | |
| 1 | 0.058 | | 0.274 | |
| 2 | 0.065 | | 0.213 | |
| Test Statistics | 5% level is 0.146 | | | |
| | 1% level is 0.216 | | | |

| Lag Order | Test Statistic D.RGDP | | Test Statistic D.RFD | |
|---|---|---|---|---|
| 0 | 0.042 | | 0.088 | |
| 1 | 0.047 | | 0.138 | |
| 2 | 0.074 | | 0.142 | |

**Table 4.** Lag Selection.

| Lag | LL | LR | df | *p* | FPE | AIC | HQIC | SBIC |
|---|---|---|---|---|---|---|---|---|
| 0 | −255.785 | | | | $2.10 \times 10^9$ | $2.71 \times 10^1$ | $2.72 \times 10^1$ | $2.72 \times 10^1$ |
| **1** | **−229.764** | **52.041** | **4** | **0.000** | **$2.10 \times 10^8$** | **24.8173** | **24.8678** | **25.1156** |
| 2 | −228.21 | 3.1081 | 4 | 0.540 | $2.70 \times 10^8$ | $2.51 \times 10^1$ | $2.52 \times 10^1$ | $2.56 \times 10^1$ |
| 3 | −226.236 | 3.9496 | 4 | 0.413 | $3.50 \times 10^8$ | $2.53 \times 10^1$ | $2.54 \times 10^1$ | $2.60 \times 10^1$ |
| 4 | −212.793 | 26.885 | 4 | 0.000 | $1.40 \times 10^8$ | $2.43 \times 10^1$ | $2.44 \times 10^1$ | $2.52 \times 10^1$ |

Note: Test Results in the Row 2 with Lag 1 shows significant result.

**Table 5.** Rank Test.

| Maximum Rank | Parms | LL | Eigenvalue | Trace Statistic | Critical Value 5% |
|---|---|---|---|---|---|
| | | | **Panel A: Lambda Trace Statistics** | | |
| 0 | 4 | −136.97855 | | 20.3449 | 12.53 |
| **1** | **7** | **−126.82926** | **0.61963** | **0.0464** | **3.84** |
| 2 | 8 | −126.80609 | 0.0022 | | |
| | | | **Panel B: Lambda Max Statistics** | | |
| 0 | 4 | −114.66567 | | 21.2573 | 11.44 |
| **1** | **7** | **−104.03703** | **0.67333** | **0.0259** | **3.84** |
| 2 | 8 | −104.02407 | 0.00136 | | |

Note: Maximum Rank is 1, highlighted in bold. This is shown in both lambda trace and lambda max.

Based on the results of Tables 4 and 5, we run the VECM model and get the estimates presented in Table 6. For _ce1.L1 we find that the estimate is 0.584 and is significant at 10% level. This is the error correction term and it is significant. The RGDP.LD coefficient is −0.685, which is also significant at 10% level. Coefficient of RFD.LD is −10.776 and significant at the 10% level. This shows that in the short-run there is relationship between the RFD and RGDP. The co-integration equation shows that the _ce1 is significant at the 1% level.

**Table 6.** VEC model estimation.

| | Coef. | Std. Err. | $z$ | $p > z$ |
|---|---|---|---|---|
| | | **D_RGDP** | | |
| _ce1 L1. | 0.584 | 0.356 | 1.640 | 0.100 |
| RGDP LD. | −0.685 | 0.400 | −1.710 | 0.087 |
| RFD LD. | −10.776 | 5.932 | −1.820 | 0.069 |
| _cons | 1.590 | 237.428 | 0.010 | 0.995 |
| | | **D_RFD** | | |
| _ce1 L1. | −0.057 | 0.019 | −3.030 | 0.002 |
| RGDP LD. | 0.031 | 0.021 | 1.450 | 0.148 |
| RFD LD. | 0.389 | 0.314 | 1.240 | 0.215 |
| _cons | 16.315 | 12.551 | 1.300 | 0.194 |

| | Cointegration Equation | | | |
|---|---|---|---|---|
| _ce1 | Params | chi2 | | $p >$ chi2 |
| | 1 | 126.969 | | 0.00 |

Identification: beta is exactly identified
Johansen normalized restrictions imposed

| | Beta | Coef. | Std. Err. | $z$ | $p > z$ |
|---|---|---|---|---|---|
| _ce1 | RGDP | 1.000 | | | |
| | RFD | 19.046 | 1.690 | 11.270 | 0.000 |
| | _cons | 148.663 | | | |

As a result, we conclude that there is long-run association between the two variables. The positive relationship between the RFD and RGDP in the regression with the Johansen normalized restrictions imposed reveals that RFD has a direct impact on RGDP, that is, the coefficient of RFD is positive and significant at the 1% level.

In Table 7, we present the results of test of autocorrelation in the error terms of VECM. Null hypothesis of the test is that there is no autocorrelation in the residuals for any order tested. At the 5% level, we cannot reject the null hypothesis. Thus, this test finds no evidence of model misspecification.

**Table 7.** Auto-Correlation Test.

| Lag | chi2 | df | Prob > chi2 |
|-----|------|----|-----|
| 1 | 2.481 | 4 | 0.648 |
| 2 | 6.686 | 4 | 0.153 |

In Table 8, we test the null hypothesis that the errors are normally distributed. In the Jarque and Bera (1980) test, we reject null for RGDP but for RFD we fail to reject the null. Overall, we reject the null. In the Skewness Test, we reject the null for RGDP but not for RFD. In the Kurtosis Test, we reject null for RGDP but not for RFD.

**Table 8.** Test of Normality of Errors.

| Equation | Skewness | chi2 | df | Prob > chi2 |
|-----|-----|-----|-----|-----|
| | | **Jarque–Bera Test** | | |
| D_RGDP | | 98.188 | 2 | 0.000 |
| D_RFD | | 0.152 | 2 | 0.927 |
| ALL | | 98.341 | 4 | 0.000 |
| | | **Skewness Test** | | |
| D_RGDP | 2.866 | 28.738 | 1 | 0.000 |
| D_RFD | −0.162 | 0.092 | 1 | 0.762 |
| ALL | 28.830 | 2.00 | 0 | |
| | | **Kurtosis Test** | | |
| D_RGDP | 11.909 | 69.450 | 1 | 0.000 |
| D_RFD | 3.263 | 0.061 | 1 | 0.806 |
| ALL | 69.511 | 2 | 0 | |

In Table 9, we do the stability test to check whether we have correctly specified the number of cointegrating equations because we specified the graph option and plotted the eigenvalues of the companion matrix. The graph of the eigenvalues shows that none of the remaining eigenvalues appears close to the unit circle. The stability check does not indicate that our model is mis-specified. Figure 1 shows a plot of unit root for the system.

**Table 9.** Test of Stability.

| Eigenvalue | | Modulus |
|-----|-----|-----|
| 1 | | 1 |
| 0.6693922 | | 0.669392 |
| −0.2323502 | +0.2016746i | 0.307667 |
| −0.2323502 | −0.2016746i | 0.307667 |

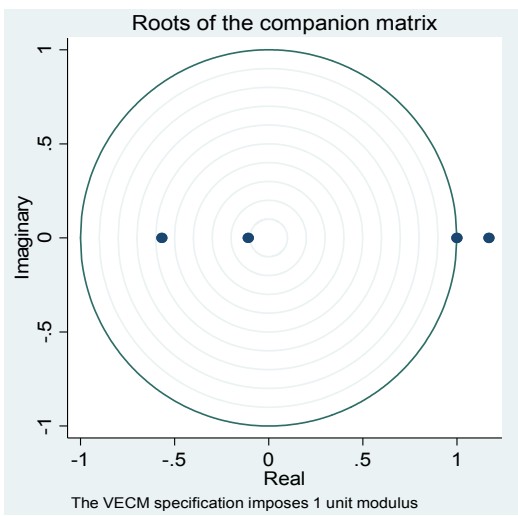

**Figure 1.** Unit-Root Circle Graph.

In the next step, we conduct the recursive cusum and cusum test of stability. Null hypothesis is that there are no structural breaks in the chosen variable. First, we do the test on GDPGR and then on FD variables. For GDPGR, we find that value of the test statistics is 0.496 which is lower than the critical values at 1, 5 and 10 percent levels of significance. The value of the test statistics for D.FD variables is 0.754. Both variables have value of the test statistics that is lower than the critical values. Thus, we fail to reject null hypothesis. Figure 2 plots the cusum square with the 95% lower and upper bound.

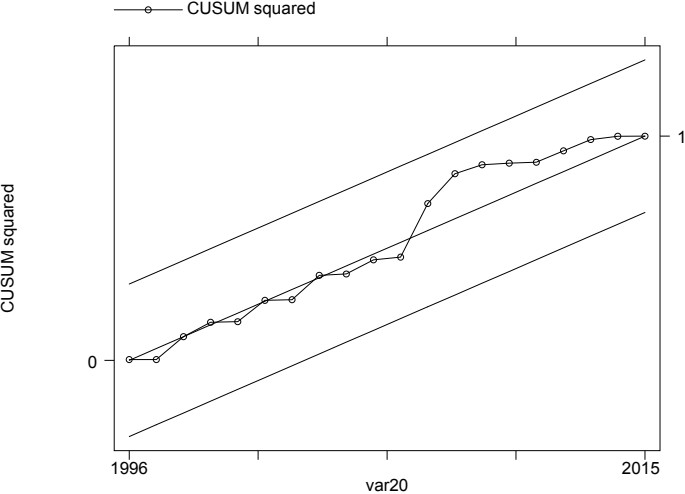

**Figure 2.** Cusum Sq Graph with 95% upper and lower bound.

In addition to the test in Table 10, we run regression of GDPGR on FD and do the cusum square test with the cumulative sum of recursive residuals and plot its value against the upper and lower bounds of the 95% confidence interval at each year. We find that the plotted line (Figure 2) fits inside the interval lines showing that there was stability during the period under consideration.

**Table 10.** Cusum and CusumSq Test of Stability.

| Variable Name | Test Statistics | Critical Value | | |
|---|---|---|---|---|
| | | 1% | 5% | 10% |
| D.GDPGR | 0.496 | 1.143 | 0.947 | 0.854 |
| D.FD | 0.754 | 1.143 | 0.947 | 0.854 |

In Table 11, we put two tests of Heteroskedasticity based on regression of GDPGR on FD. Null hypothesis is that all variances are equal. We fail to reject null in both tests. We do not find evidence of heteroskedasticity.

**Table 11.** Test of Heteroskedasticity.

| | *F*-Stat | Prob > *F* |
|---|---|---|
| Breusch–Pagan Test | $F(1, 21) = 2.75$ | 0.1119 |
| The White Test | $F(2, 20) = 1.31$ | 0.2915 |
| | **LM-Statistics** | **Chi-Square Critical Value of 5% Level of Significance** |
| Breusch–Pagan Test | $23 \times 0.116 = 2.668$ | 32.671 |
| The White Test | $23 \times 0.116 = 2.668$ | 33.924 |

*4.1. Additional Discussion and Results*

In the appendix of the study, we present the findings from similar analysis VECM models based on data for the World Bank covering the period 2001–2014. Fiscal Deficit (FD) and annual growth rate of GDP (GDPGR). In Table A2 of the appendix, we see that the two variables have a correlation coefficient of 0.360. Table A3 Panel A in the appendix shows that Dickey–Fuller tests show that there is no unit root in FD series at the 5% or 10% levels, but the GDPGR series is non-stationary. The first difference of the series is stationary at the 5% or 10% level. Panels B and C of Table A3 show results of Phillips–Perron and KPSS tests. Table A4 of the appendix shows that most of the criteria indicate that the lag selected should be two. In Table A5, we do the rank test and find the rank to be of order one, which indicates that we can do the VECM test. Table A6 presents the VECM result. The _ce1.L1 in the D_GDPGR equation shows that the item has a significant and negative coefficient at the 1% level. Thus, error-correction term plays an important role in determining D_GDPGR. In the D_FD equation, the _ce1.L1 has a negative and insignificant coefficient. The GDPGR.LD has a positive and insignificant coefficient. It is evident that these variables have strong impact on the D_FD variable in the short run. In the bottom part of the above Table A6, the cointegration equation is presented. We find that the coefficient of the FD variable is small but negative and significant at the 5% level. As a result, we conclude that the impact of Fiscal Deficit (FD) on GDPGR is mild but negative, that is, a larger fiscal deficit slows down the rate of growth of the economy. This is contrary to Keynesian theory, but is in conformity with Neo-classical theory, which holds that fiscal deficits lead to a fall in the Gross Domestic Product. If deficit spending has a negative effect, it may be due to its effect on raising the long-term interest rate. Table A7 shows the result for test of auto-correlation in error terms. At the 5% level, we cannot reject the null hypothesis. The results clearly indicate that there is serial correlation in the residuals. Table A8 shows normality test of error terms. We reject the null that the error term of the VECM model is normally distributed. Most of the errors are not skewed and kurtotic. Thus, the errors in Table A9 show stability of the system. We find that only one of the Eigenvalue modulus falls outside the unit circle, which indicates that the system is mostly stable. These findings also indicate that the model is not mis-specified. In Figure A1 we find that only none of the Eigen value modulus fall out-side the unit circle, which indicates that the system is mostly stable. These findings also indicate that the model is not mis-specified. The cusum tests also show that the variables are stable. Figure A2 of the cusum square shows that the line does not cross the 95% upper and lower bounds. In Table A10, test of heteroskedasticity shows evidence of homoscedasticity.

## 5. Conclusions

Our study researches the relationship between fiscal deficit and its impact on economic growth in Bangladesh, using two different datasets from two different sources. The Bangladesh Bureau of Statistics' (BBS) dataset is the local source, while the World Bank Development Indicators (WBDI) dataset from World Bank is the foreign source. The findings from the VECM for BBS data reveals that there is a positive and significant relationship between FD and GDPGR, supporting the Keynesian theory, while from the VECM for World Bank data indicates that the impact of Fiscal Deficit (FD) on GDPGR is mild but negative and significant at the 5% level. This contradicts the Keynesian theory, but is in accord with Neo-classical theory which asserts that fiscal deficits lead to a drop in the GDP. Using two different datasets in the same study is quite distinctive and based on our findings in the context of Bangladesh, we think researchers from different countries will try to replicate our approach and contribute to the literature. Given our results, we emphasize that quality of government expenditure is important and major emphasis should be given to make sure that government expenditure should be undertaken after careful planning. It should be effectively implemented so that benefits of such projects are substantial. Delays in implementation of such projects and cost-overruns should be brought down to a minimum. Government should give priority to projects that deal with public goods and positive externality. The private sector alone will not provide enough of such goods, some of which are essential to national development. Nevertheless, the government must keep the deficit under control so that the growth of the economy may continue without any pause. Fiscal policy has an important role on growth of the economy in Bangladesh; therefore, it is imperative on Government's part to carefully formulate the tax policy (to generate revenue) and expenditure policy (checking wasteful expenditure and curbing corruption) keeping in mind that such policies have significant impact on the growth. Fiscal policy adjustment that reduces unproductive expenditure and protects expenditure in the social sector, can lead to additional sustainable level and is likely to result in faster growth. The level of government expenditure should be set so as to avoid huge deficits leading to debt financing and the crowding-out effect of private investment. If deficits become unsustainable, it can lead to higher interest payments, loss of confidence and lower GDP growth rate. To decrease the fiscal deficit, the government can raise the taxes and cut government spending, but this may perhaps lower economic growth. Intent of the tax policy needs to focus on a well-planned basis and be executed in combination with tax incentives and tax holidays for businesses to flourish. Given the low level of savings and investable funds, as is typical in a developing country like Bangladesh, the need to mobilize more resources (both foreign and domestic) to invest in productive sectors of the economy can hardly be overemphasized.

Given the finding from this study, on the relationship between fiscal deficit and economic growth in Bangladesh, policy makers have a duty to ensure that most of the expenditure outlay is fully, timely and effectively utilized. Emphasis has to be given to increase capital formation (in the government outlay) while keeping the current consumption of the Government under check. The policy makers, donors, multinational organizations, development partners in Bangladesh should all put emphasis on this "Golden Rule" of public finance for now.

Although in the economic literature, there is no definitive conclusion as to whether fiscal deficit helps or hinders economic growth for any country, many argue that fiscal deficit leads to economic growth of a country, which cannot be achieved only through domestic savings, not enough for investment. It can be assumed safely that a certain amount of fiscal deficit is good for economic growth, if the borrowed money is expended for beneficial proposes, provided the return from such investments exceeds the funding cost. For further research works on Bangladesh's economic growth, it will be interesting to examine the relationship between government spending and long-term interest rate, inflation rate, exchange rate, trade deficit, private investment, poverty alleviation for the period from 2001 onwards as the results from WB data indicated a mild negative relationship between fiscal deficit and GDP growth.

# Appendix A

**Table A1.** Summary Statistics.

|  | Obs | Mean | Std. Dev. |
|---|---|---|---|
| GDPGR | 13 | 5.75 | 0.93 |
| FD | 13 | 293692.5 | 206832.9 |

**Table A2.** Correlation.

|  | GDPGR | FD |
|---|---|---|
| GDPGR | 1 |  |
| FD | 0.360 | 1 |

**Table A3.** Unit root and Stability Tests.

| | Panel A: Dickey Fuller Test | | | |
|---|---|---|---|---|
| **Z(t)** | **Test Statistics** | **1% Critical Value** | **5% Critical Value** | **10% Critical Value** |
| GDPGR | −1.628 | −3.75 | −3.00 | −2.63 |
| D.GDPGR | −3.148 | −3.75 | −3.00 | −2.63 |
| FD | 3.355 | −3.75 | −3.00 | −2.63 |
| | Panel B: Phillips-Perron Test | | | |
| | **Test Statistics** | **1% Critical Value** | **5% Critical Value** | **10% Critical Value** |
| | **GDPGR** | | | |
| Z(rho) | −6.872 | −17.200 | −12.500 | −10.200 |
| Z(t) | −1.982 | −3.750 | −3.000 | −2.630 |
| | **FD** | | | |
| Z(rho) | 2.567 | −17.200 | −12.500 | −10.200 |
| Z(t) | 4.499 | −3.750 | −3.000 | −2.630 |
| | **D.GDPGR** | | | |
| Z(rho) | −15.726 | −17.200 | −12.500 | −10.200 |
| Z(t) | −3.852 | −3.750 | −3.000 | −2.630 |
| | **D.FD** | | | |
| Z(rho) | −3.382 | −17.200 | −12.500 | −10.200 |
| Z(t) | −1.369 | −3.750 | −3.000 | −2.630 |
| | Panel C: KPSS Tests | | | |
| Lag Order | Test Statistic GDPGR | | Test Statistic RFD | |
| 0 | 0.103 | | 0.313 | |
| 1 | 0.072 | | 0.189 | |
| 2 | 0.069 | | 0.155 | |
| Test Statistics | 5% level is 0.146 | | | |
| | 1% level is 0.216 | | | |
| Lag Order | Test statistic D.GDPGR | | Test statistic D.RFD | |
| 0 | 0.0483 | | 0.0665 | |
| 1 | 0.0475 | | 0.0764 | |
| 2 | 0.0559 | | 0.134 | |

Note: Test Results in the Row 3 with Lag 2 shows significant result.

**Table A4.** Lag Selection Criteria.

| lag | LL | LR | df | *p* | FPE | AIC | HQIC | SBIC |
|---|---|---|---|---|---|---|---|---|
| 0 | −7.89036 | | | | 0.422604 | 1.97564 | 1.92835 | 1.99755 |
| 1 | −7.57938 | 0.62195 | 1 | 0.430 | 0.495799 | 2.12875 | 2.03417 | 2.17258 |
| **2** | **−3.44336** | **8.2721 \*** | **1** | **0.004** | **0.251695** | **1.43186** | **1.28999** | **1.4976** |
| 3 | −2.76117 | 1.3644 | 1 | 0.243 | 0.281176 | 1.50248 | 1.31332 | 1.59014 |
| 4 | −2.75633 | 0.00968 | 1 | 0.922 | 0.3781 | 1.72363 | 1.48718 | 1.8332 |

Note: Test Results in the Row 3 with Lag 2 shows significant result.

**Table A5.** Rank Selection/Tests.

| Maximum Rank | Parms | LL | Eigenvalue | Trace Statistic | Critical Value 5% |
|---|---|---|---|---|---|
| | | | **Lambda Trace** | | |
| 0 | 6 | −141.16117 | | 15.5272 | 15.41 |
| **1** | **9** | **−134.47017** | **0.70375** | **2.1452 *** | **3.76** |
| 2 | 10 | −133.39756 | 0.17718 | | |
| | | | **Lambda Max** | | |
| 0 | 6 | −293.13178 | | 13.382 | 14.07 |
| 1 | 9 | −286.44078 | 0.70375 | 2.1452 | 3.76 |
| 2 | 10 | −285.36818 | 0.17718 | | |

Note: Test Results in the Row 2 with Lag 1 shows significant result.

**Table A6.** Vector Error Correction Models.

| | Coefficient | Standard Deviation | *t*-stat | *p*-value |
|---|---|---|---|---|
| | | **D_GDPGR** | | |
| _ce1 L1. | −0.554 | 0.136 | −4.060 | 0.000 |
| GDPGR LD. | 0.276 | 0.170 | 1.620 | 0.105 |
| FD LD. | 0.000 | 0.000 | −4.860 | 0.000 |
| _cons | −2.446 | 0.762 | −3.210 | 0.001 |
| | | **D_FD** | | |
| _ce1 L1. | −3998.416 | 13545.850 | −0.300 | 0.768 |
| GDPGR LD. | 14113.690 | 16862.940 | 0.840 | 0.403 |
| FD LD. | 0.536 | 0.430 | 1.250 | 0.213 |
| _cons | 0.000 | 75680.910 | 0.000 | 1.000 |

| | Cointegrating equations | | | |
|---|---|---|---|---|
| _ce1 | Parms | | chi2 | *p* > chi2 |
| | 1 | | 4.663 | 0.031 |

Identification: beta is exactly identified
Johansen normalization restriction imposed

| | beta | Coef. | Std.Err | Z | *p* > z |
|---|---|---|---|---|---|
| _ce1 | GDPGR | 1 | | | |
| | FD | $-9.97 \times 10^{-6}$ | $4.62 \times 10^{-6}$ | −2.16 | 0.031 |
| | _cons | −9.632928 | | | |

**Table A7.** Autocorrelation. Lagrange Multiplier test.

| lag | chi2 | df | *p*−Value |
|---|---|---|---|
| 1 | 2.439 | 4 | 0.656 |
| 2 | 2.063 | 4 | 0.724 |

**Table A8.** Normality Tests.

| Jarque-Bera test | | | |
|---|---|---|---|
| **Equation** | **chi2** | **df** | ***p*-Value** |
| D_GDPGR | 0.294 | 2 | 0.863 |
| D_FD | 0.610 | 2 | 0.737 |
| ALL | 0.904 | 4 | 0.924 |
| **Skewness test** | | | |
| **Equation** | **Skewness** | **chi2** | **df** | ***p*-Value** |
| D_GDPGR | −0.353 | 0.228 | 1 | 0.633 |
| D_FD | 0.080 | 0.012 | 1 | 0.914 |
| ALL | | 0.240 | 2 | 0.887 |
| **Kurtosis test** | | | |
| **Equation** | **Kurtosis** | **chi2** | **df** | ***p*-Value** |
| D_GDPGR | 2.622 | 0.066 | 1 | 0.798 |
| D_FD | 1.858 | 0.598 | 1 | 0.439 |
| ALL | | 0.664 | 2 | 0.718 |

**Table A9.** Stability Tests.

| Panel A: Unit Modulai Test. | | |
|---|---|---|
| **Eigenvalue** | | **Modulus** |
| 1 | | 1 |
| 0.9410217 | | 0.941022 |
| 0.1782671 | +0.6641845i | 0.687692 |
| 0.1782671 | −0.6641845i | 0.687692 |

| Panel B: Cusum Tests | | | | |
|---|---|---|---|---|
| | | **Critical Value** | | |
| **Variable Name** | **Test Statistics** | **1%** | **5%** | **10%** |
| D.RGDP | 0.457 | 1.143 | 0.948 | 0.850 |
| D.RFD | 0.893 | 1.143 | 0.948 | 0.850 |

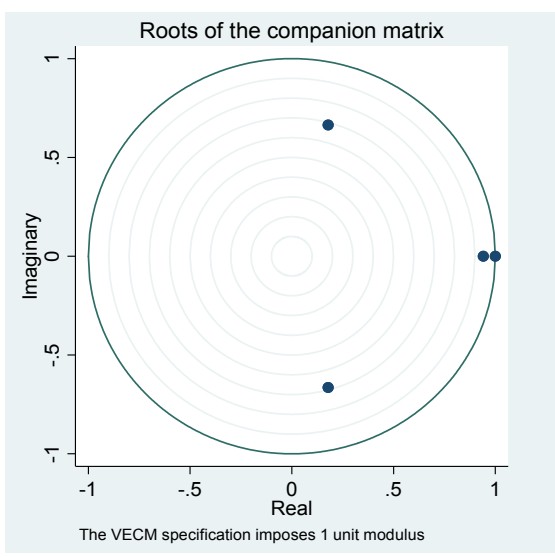

**Figure A1.** Unit Root Stability Check.

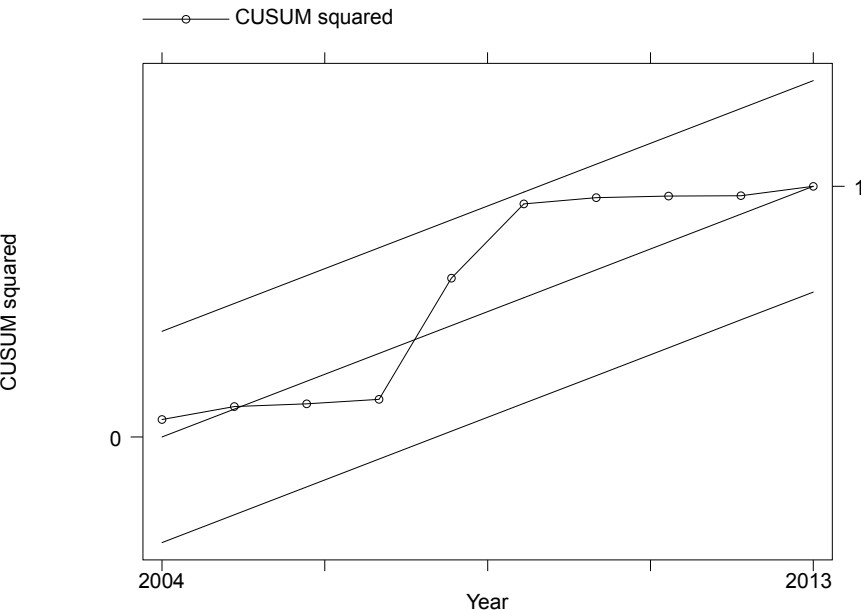

**Figure A2.** Cusum Stability Check.

**Table A10.** Heteroskedasticity Tests.

|  | *F*-Stat | **Prob > *F*** |
|---|---|---|
| Breusch-Pagan Test | 1.24 | 0.312 |
| The White Test | 1.46 | 0.278 |
|  | LM-statistics | Prob > chi2 |
| Breusch-Pagan Test | 1.940 | 0.163 |
| The White Test | 2.930 | 0.231 |

**Author Contributions:** Both authors contributed equally to this work.

**Conflicts of Interest:** The authors declare no conflict of interest.

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
