# Peer review of "Fiscal Deficit and Its Impact on Economic Growth: Evidence from Bangladesh"

_economies, doi:10.3390/economies5040037_

Round 1
Reviewer 1 Report
Dear Authors,
I think that the paper raises a very interesting and topical subject. However several changes have to be done in order to be published.
The paper investigates the impact of Fiscal Deficit on Economic in Bangladesh using data from two sources (two data-sets, BBS and WBDI database). Since the number of observations is limited (1993-2016 for BBS) it is good that the authors use a similar analysis based on the World Bank data (2001-2014) in order to ensure/compare their results.
Generally we can say that the research has been described appropriate. Robust methodology steps are applied. Nevertheless, additional methods/test has to be done. Also, some of the methods that appear in the paper have to be corrected.
Finally, the conclusions presented are supported by the results. However, as I mention above, some econometric estimations have to be added/corrected. It is worth to say that all the results are clearly presented (can be improved).

Author Response
Referee # 1.
Reply to the Referee’s Comments. We wish to thank the anonymous referee for the review of our manuscript and for the valuable suggestions/comments and feedbacks. We have followed it very closely while revising the draft and believe the manuscript to be a much better product as an effect. We chronicle below our response and the revisions that have been made to the paper. We sincerely hope that the Editor and the Referees will be satisfied with our revised work done in Round 2.
“I think that the paper raises a very interesting and topical subject. However several changes have to be done in order to be published. The paper investigates the impact of Fiscal Deficit on Economic in Bangladesh using data from two sources (two data-sets, BBS and WBDI database). Since the number of observations is limited (1993-2016 for BBS) it is good that the authors use a similar analysis based on the World Bank data (2001-2014) in order to ensure/compare their results. Generally we can say that the research has been described appropriate. Robust methodology steps are applied. Nevertheless, additional methods/test has to be done. Also, some of the methods that appear in the paper have to be corrected. Finally, the conclusions presented are supported by the results. However, as I mention above, some econometric estimations have to be added/corrected. It is worth to say that all the results are clearly presented (can be improved)”.
General Comments
1. Can the authors stress some aspects making their work innovative?
2. The paper has to be proofread by a native speaker so some syntax errors to be corrected.
We have acted accordingly on the valuable advice. Regarding suggestion for “stress some aspects making their work innovative”, we have put greater emphasis throughout the draft to point this out. “Our study researches the relationship between fiscal deficit and its impact on Economic Growth in Bangladesh, using two different data set from two different sources. Bangladesh Bureau of Statistics’ (BBS) data set is the local source while the World Bank Development Indicators (WBDI) data set from World Bank is the foreign source”. This is quite distinct by itself. Our sincere thanks to the anonymous Referee.
Theoretical Comments
3. The paper investigates the impact of Fiscal Deficit on Economic in Bangladesh using data from two sources (two data-sets, BBS and WBDI database). Since the number of observations is limited (1993-2016 for BBS) it is good that the authors use a similar analysis based on the World Bank data (2001-2014) in order to ensure/compare their results. No action required.
4. I recommend the authors to separate the literature section in theoretical and empirical. Expected effects should be underpinned by theoretical models. Also, in theoretical section the three schools of economic though (Neoclassical, Keynesian, Ricardian Equivalence (transfer it from introduction section)) regarding the impact of budget deficits and recent theoretical studies can be added. We have reorganized the Introduction and Lit Review section upon the advice given by the Referee. Thank you!
5. I recommend the authors to add in introduction section a few words about the impact of global financial crisis on the economy of Bangladesh and the factors that have helped the country to survive the global challenges. Thanks to the Referee for the great suggestion. “The financial crisis of 2007-2008 in United States was followed by a global economic meltdown that led many to believe that Bangladesh economy too will be adversely affected. But, Bangladesh came out doing fairly well. According to the study by Ali et.al. (2011), the authors find that the macro economy of Bangladesh has shown remarkable resilience in the face of this massive global crisis, and the impact has been minimal and limited to a moderate slowdown of the economy”. We have also added a footnote # 5 in the main draft a recent report from World Bank on Bangladesh economy. Thanks!
6. In the empirical literature section, the review should also provide more details on the methodology employed by the relevant studies to verify comparability of the results discussed. Also, it would be better if the authors concentrate only on evidence from studies that concern developing countries as the paper is conducted on such country. It is good that the most studies mentioned are from recent empirical literature. We have followed the advice given by the Referee and have added the in the cited studies the methodology followed the authors/s in their study. Once again many thanks.
Methodology Comments
7. The authors begin their analysis applying the ADF unit root test in order to define the stationarity properties of the variables. Except form ADF I recommend the use of PP test as well as the KPSS test. We thank the Referee for the suggestion. Please see our action taken. In Table 3 panel B we added results for Phillips-Perron Tests and in panel C we added the results of the KPSS tests. For the model in the appendix, we placed the result in Table 14. Thank you!
8. Having defined the integration order of the series they continue applying the Johansen and Juselius (1990) cointegration approach in order to examine the long run relationship among the variables. Since this approach is sensitive to the lag length, before applying the cointegration test, they rightly found the order of the VECM (by the minimum value of the Akaike information criterion (AIC), Schwarz information criterion (SBC) and Hannan-Quinn criterion (HQC). As suggested by the Referee, the following actions were taken. In Table 5 we presented the result for Rank test and in Table 4 we present selection of lag-order for the VECM model. For the model in the appendix Table 14 and 15 shows the results for similar tests. Thank you!
9. At line 241 (page 6) it would be good to mention that the FMOLS approach for cointegration is used only in panel data. We have added that “FMOLS approach is only used in panel data”. In page 12. This correction is at the bottom of page #11 in the revised draft. Once again our sincere thanks.
10. The VECM equations have to be written again as follows (estimations to be done again, Table 6): Please see our action taken in this regard. The equations were written in the format suggested by the referee. In the revised draft they appear on page 12. Thank you!
Empirical Results Comments.
11. Unit root results: The corrections were carried out in the discussion section of the draft and in the Unit Root test results for Table 3. Thanks for the valuable suggestions.
12 Cointegration results: At line 283-284 “in Table 5 we present the results of trace test”. Why the authors don’t mention about the λ max tests? Thank you for the great suggestion. We added the lambda (λ) max statistics in Table 5 panel B for the BBS series. For the WB bank series, we presented the statistics in Table 15 panel B.
13. The estimates obtained should be tested for robustness before any sensible conclusions are drawn. So in addition to authors’ diagnostic tests the White and the LM test (Heteroskedasticity) have to be added.
Action taken as advised. We added the tests for heteroskedasity: the Breusch-Pagan Test and the White test with both LM statistics and F-tests. Table 11 shows these results. For the model in the appendix Table 22 shows the tests. Thank you!
14. As far as the stability test I propose the authors to appear the (inverse) unit root circle graph. In addition, the Cusum and the Cusum test of squares can be added. Acted as advised. We added the unit root circle for stability test in Fig-1 and cusum square test in Fig 2, respectively. In Table 10, we present the cusum test of structural break on individual variables. We find that there is no structural breaks in the variables. Fig 2 shows the cusum square with a 95 percent upper and lower bound. This test also shows that the model is stable. Fig 3 and 4 presented unit-root graph and cusum square graph with 95 percent bounds for the model in the appendix, respectively. The 95 percent upper and lower bound also show that the system is stable. Table 20 panel B shows cusum tests for the model presented in the appendix.
Minor Comments
1. Reference Style In page 6 at section Data, Bangladesh and Bureau of Statistics (BBS, Year?) and World Bank Development (WBDI, Year?). We added the source of BBS data as: BBS 2017. We also added the World Development Indicator WDI online as: WDI online 2017. Reference section was also corrected. Once again our sincere thanks to the anonymous Referee. We believe, the valued suggestions made by the Referee has helped us vastly improve the draft in this round. Thank you!
Reviewer 2 Report
I would like to thank the authors for their research within this area, which is of much interest to me. In this way, Researcher(s) grounded the study on the fiscal depict and economic growth in Bangladesh perspective. Quick google scholar search indicated that, no of studies were grounded in this title including in the Bangladesh context. I am struggling to identify what is new about this research. The way of writings makes bold claims without really highlighting what’s new in this research, I would reduce these claims as previous research has looked quite extensively at this area.
Other than the novelty, Author(s) constructed the research article in an acceptable standard. Author(s) documented some key facts about the economic condition of the Bangladesh especially in the introduction part. Meanwhile, research question and problem statement were not adequately pinpointed. Therefore, Author(s) should focus on that area tactfully. Interestingly, Researcher(s) also critically reviewed and documented the various theoretical underpinnings and recent empirical works. However, they didn’t note what extent the present study differs from the extant literature.
Researcher(s) properly utilized the techniques to answer the main research questions and validated the basic criteria in econometrics approach such as Unit root Test. To this end, Researcher(s) systematically discussed the findings. However, researcher(s) didn’t discuss about the implication for theory and practice, limitation and further research direction. Therefore, I kindly request the researchers to incorporate the implication for theory and practice, in which author(s) should note down the facts what extent the findings of the study are consistent with the extant theoretical underpinnings (e.g. Keynesian theory). Moreover, they also suggest the future researchers to drilldown the relationship between fiscal deficit and economic growth. For example, researcher(s) may find the different mechanisms (moderator or mediator variable)to link the fiscal deficit and economic growth.
Author Response
All our rejoinder/s are in bold.
Referee # 2
Reply to the Referee’s Comments. We wish to thank the anonymous referee for the review of our manuscript and for the valuable suggestions/comments and feedbacks. We have followed it very closely while revising the draft and believe the manuscript to be a much better product as an effect. We chronicle below our response and the revisions that have been made to the paper. We sincerely hope that the Editor and the Referees will be satisfied with our revised work done in Round 2.
“I would like to thank the authors for their research within this area, which is of much interest to me. In this way, Researcher(s) grounded the study on the fiscal depict and economic growth in Bangladesh perspective. Quick google scholar search indicated that, no of studies were grounded in this title including in the Bangladesh context. I am struggling to identify what is new about this research. The way of writings makes bold claims without really highlighting what’s new in this research, I would reduce these claims as previous research has looked quite extensively at this area. Other than the novelty, Author(s) constructed the research article in an acceptable standard. Author(s) documented some key facts about the economic condition of the Bangladesh especially in the introduction part. Meanwhile, research question and problem statement were not adequately pinpointed. Therefore, Author(s) should focus on that area tactfully. Interestingly, Researcher(s) also critically reviewed and documented the various theoretical underpinnings and recent empirical works. However, they didn’t note what extent the present study differs from the extant literature”.
We thank the referee for his/her valuable comments. Our present study is different from previous study in several key areas: we use two data-sets from two sources, one is the official source of Bangladesh Government and the other is the World Bank online data-base. We collected the longest series that was available. Another important area covered in the present study is that we have conducted array of tests in every stages often checking the result of one test with another, for example, we conduct three different approaches to unit-root tests, for example, Dicky-Fuller Tests, Phillips-Perron Tests, and KPSS tests, two versions of structural break tests: one, cusum and cusum square; and two, cusum square 95 percent bound test, , rank test with lamda max statistics, the Breusch-Pagan Test and the White test of Homoscedasticity. All these tests were added following the recommendation of another anonymous Referee.
“Researcher(s) properly utilized the techniques to answer the main research questions and validated the basic criteria in econometrics approach such as Unit root Test. To this end, Researcher(s) systematically discussed the findings. However, researcher(s) didn’t discuss about the implication for theory and practice, limitation and further research direction. Therefore, I kindly request the researchers to incorporate the implication for theory and practice, in which author(s) should note down the facts what extent the findings of the study are consistent with the extant theoretical underpinnings (e.g. Keynesian theory). Moreover, they also suggest the future researchers to drilldown the relationship between fiscal deficit and economic growth. For example, researcher(s) may find the different mechanisms (moderator or mediator variable) to link the fiscal deficit and economic growth”.
Our response: In this research, we find that there is a long term relationship between GDP growth rate and government budget deficit. This supports Keynesian theory, who argued that the economy by itself will not return to equilibrium due to price rigidity. As a result, the government should take steps to fine-tune public investment and change aggregate demand to stimulate GDP growth rate. Our findings in this paper opposes Monetarist point of view, who believes that only monetary policy matters but fiscal policy does not matter for GDP growth rate. We find that fiscal policy working through government budget deficit does have important impact on Bangladesh economy. Centered on this findings, we recommend that the government policy makers and the donor agencies, and the private sector policy makers all put greater importance on the government budget deficit. The government needs to continue to invest in the economy to stimulate growth. We have highlighted this point of importance in the conclusion section of the paper.
One limitation of our study is that we have a short data-period, which may not be suitable to conduct the sophisticated econometric techniques that we have used in the paper. To address the problem, we have used two data-sets instead of one and subjected both to the same vigorous statistical analysis. We used the longest possible data-sets available from BSS and the World Development Indicators. We placed the statistical results of the BSS in the body of the tests and that of the World Development Indicators in the appendix. In the conclusion section of the paper, we have presented some avenues of future research.
Reviewer 3 Report
The empirical analysis is very poor and completely unreliable. Unit root tests and cointegration methods need large samples to provide reliable inference. The author/s employ a very limited number of observations for this kind of analysis which leads to completely unreliable statistical inference. The use of a system approach (Johansen test) further deteriorates the problem because the number of degrees of freedom is considerably lower than it should be. And, consequently, all economic conclusions and policy implications could be misleading as well.
Author Response
All our rejoinder/s are in bold.
Referee # 3.
Reply to the Referee’s Comments. We wish to thank the anonymous referee for the review of our manuscript and for the valuable suggestions/comments and feedbacks. We have followed it very closely while revising the draft and believe the manuscript to be a much better product as an effect. We chronicle below our response and the revisions that have been made to the paper. We sincerely hope that the Editor and the Referees will be satisfied with our revised work done in Round 2.
“The empirical analysis is very poor and completely unreliable. Unit root tests and cointegration methods need large samples to provide reliable inference. The author/s employ a very limited number of observations for this kind of analysis which leads to completely unreliable statistical inference. The use of a system approach (Johansen test) further deteriorates the problem because the number of degrees of freedom is considerably lower than it should be. And, consequently, all economic conclusions and policy implications could be misleading as well”.
We thank you for rightly pointing out that both the data-sets (the BBS and the World Bank) are short. To address the problem of limited data-sets we have used to separate sets (the BBS and the World Bank) to examine the relationship between fiscal deficit and economic growth rate. They give comparable results. At the same time, we implemented new tests of unit-root, KPSS, Phillips-Perron, cusum and cusum square tests etc. following the suggestions of Referee 1 to enrich our result. When we compare the result of one test with the result of another test, they should give us better understanding and validity. We also like to state that our endeavor in this study was limited by data-availability constraint arising from data collected from the official source. We can only use data that the official sources makes it available. We tried to collect as many years of data as we could. However, we took the longest data-set we could find form both the BBS and the World Bank’s Development Indicators.